# Repeated Vaginal Exposures to the Common Cosmetic and Household Preservative Methylisothiazolinone Induce Persistent, Mast Cell-Dependent Genital Pain in ND4 Mice

**DOI:** 10.3390/ijms20215361

**Published:** 2019-10-28

**Authors:** Erica Arriaga-Gomez, Jaclyn Kline, Elizabeth Emanuel, Nefeli Neamonitaki, Tenzin Yangdon, Hayley Zacheis, Dogukan Pasha, Jinyoung Lim, Susan Bush, Beebie Boo, Hanna Mengistu, Ruby Kinnamon, Robin Shields-Cutler, Elizabeth Wattenberg, Devavani Chatterjea

**Affiliations:** 1Biology Department, Macalester College, Saint Paul, MN 55105, USA; arriagagomez.erica@gmail.com (E.A.-G.); jaclynmlkline@gmail.com (J.K.); elizabethe819@gmail.com (E.E.); nefnea@gmail.com (N.N.); tyangdon@macalester.edu (T.Y.); hzacheis@macalester.edu (H.Z.); dpasha@macalester.edu (D.P.); beebieboo@gmail.com (B.B.); hmengist@umn.edu (H.M.); rkinnamo@macalester.edu (R.K.); rshield2@macalester.edu (R.S.-C.); 2Mathematics, Statistics & Computer Science Department, Macalester College, Saint Paul, MN 55105, USA; jlim2@macalester.edu; 3Biology Department, Trinity College, Hartford, CT 06106, USA; susan.bush@trincoll.edu; 4Division of Environmental Health Sciences, University of Minnesota School of Public Health, Minneapolis, MN 55455, USA; watte004@umn.edu

**Keywords:** mast cells, allergy, vulvar pain, methylisothiazolinone, Δ-9 tetrahydrocannabinol

## Abstract

A history of allergies doubles the risk of vulvodynia—a chronic pain condition of unknown etiology often accompanied by increases in numbers of vulvar mast cells. We previously established the biological plausibility of this relationship in mouse models where repeated exposures to the allergens oxazolone or dinitrofluorobenzene on the labiar skin or inside the vaginal canal of ND4 Swiss Webster outbred mice led to persistent tactile sensitivity and local increases in mast cells. In these models, depletion of mast cells alleviated pain. While exposure to cleaning chemicals has been connected to elevated vulvodynia risk, no single agent has been linked to adverse outcomes. We sensitized female mice to methylisothiazolinone (MI)—a biocide preservative ubiquitous in cosmetics and cleaners—dissolved in saline on their flanks, and subsequently challenged them with MI or saline for ten consecutive days in the vaginal canal. MI-challenged mice developed persistent tactile sensitivity, increased vaginal mast cells and eosinophils, and had higher serum Immunoglobulin E. Therapeutic and preventive intra-vaginal administration of Δ^9^-tetrahydrocannabinol reduced mast cell accumulation and tactile sensitivity. MI is known to cause skin and airway irritation in humans, and here we provide the first pre-clinical evidence that repeated MI exposures can also provoke allergy-driven genital pain.

## 1. Introduction

Debilitating, unexplained provoked localized vulvar pain or vulvodynia affects a significant proportion (~8%) of women [1,2] and is epidemiologically linked to a history of both seasonal and contact allergies [3]. Vulvar biopsies from diagnosed patients show increases in mast cells and nerves [4,5,6]. To recapitulate and dissect the pathobiology of allergy-driven tactile sensitivity and to inform novel therapies, we established mouse models of allergy-driven genital pain [7,8,9]. We have shown that contact hypersensitivity to the commonly used laboratory haptens oxazolone (Ox) on the labiar skin or dinitrofluorobenzene (DNFB) on the labiar skin or vaginal canal induces persistent tactile genital pain and increased accumulation of mast cells in the labiar tissues of outbred, female ND4 mice well beyond the resolution of visible inflammation. Chemical depletion of labiar mast cells reduced Ox-driven painful responses [8] and therapeutic topical administration of Δ-9-tetrahydrocannabinol (THC) in the vaginal canal alleviated DNFB-induced pain and reduced numbers of accumulated mast cells in the affected tissue [9]. Here, we examined the potential of a common household chemical, 2-methyl-4-isothiazolin-3-one/methylisothiazolinone (MI), to induce contact hypersensitivity reactions and consequent allergy-driven genital pain in ND4 female outbred mice. MI is a biocide preservative present in soaps, shampoos, vaginal washes, household cleaners, and paints [10,11]. Recent evidence suggests that a significant portion of people exposed to MI developed the capacity for an allergic response [11,12], showed exacerbated inflammatory responses [13], and experienced tissue injury in the skin or lungs [12,14] after exposure. Allergic responses to MI have also been linked to vulvar dermatitis [15,16], but no connections between such dermatoses and the later development of vulvar pain have been made in the published clinical literature. Recently, Reed and colleagues identified exposures to household and workplace chemicals as a possible risk factor for the development of vulvodynia [17]. Our previously published mouse models demonstrated the biological plausibility of the epidemiological link between chemical exposures and the development of genital pain. However, a specific link between vulvodynia and a known environmental chemical does not yet exist. We suggest that MI is a plausible candidate for an environmental irritant/allergen, and that exposure to MI might drive allergy-provoked pain. Here, we repeatedly applied MI dissolved in saline (a surrogate for water-based cleansers that typically contain MI as a preservative) topically within the vaginal canals of mice and characterized consequent allergic inflammation, accumulation of mast cells, and ano-genital sensitivity to pressure. We also assessed the effects of therapeutic and preventive administration of topical THC treatments in the vaginal canal on mast cell abundance and painful sensitivity.

## 2. Results

### 2.1. Meta-Analysis of Patch-Testing Studies Reveals Widespread Sensitization to MI in Populations Tested in Europe and North America

MI has been in use as a biocide preservative for many decades by itself and, earlier, in conjunction with methylchloroisothiazolinone (MCI), and adverse allergic reactions to these chemicals were reported as early as the 1980s [18,19] in both clinical and pre-clinical studies [20,21]. MCI was considered a strong sensitizer and subsequently discontinued; MI, which was previously used at a lower concentration in the MI/MCI mixture, was used at a higher concentration by itself, and safe limits for MI in cosmetic preparations were set at 100 ppm [22]. However, reports of allergic outcomes as well as sensitization of populations exposed to the purported safe doses of MI via household and industrial contexts have continued to rise. We conducted a meta-analysis of studies that reported MI-sensitivity via patch testing to evaluate the epidemiological trends of allergic sensitivity to MI. Of the 163 epidemiological studies, between 1995 and 2019, reported on PubMed.gov, most were conducted in Europe, along with a few North American investigations (Figure 1A). One article without MI parts per million (ppm) tested was excluded from plotting [23]. Sample sizes varied between fewer than 100 and more than 10,000 human subjects and concentrations of MI used varied between 500 and 2000 ppm. By 2012, studies reporting 5–10% of participants being MI-sensitive were more prevalent, while the majority of studies reporting 1% of participants sensitized to MI were published prior to 2012 (Figure 1B). In 2013, the American Contact Dermatitis Society named MI the Allergen of the Year [24]. Sensitivity to MI is clear and present in populations within North America and Europe, and adverse health outcomes related to MI are on the rise. Given that exposure to household cleaning products is a risk factor for developing vulvodynia [17], we found it important to investigate whether repeated exposure to MI provoked persistent pain in laboratory mice.

### 2.2. Repeated Exposures to MI in the Vaginal Canal Induce Painful Ano-Genital Responses to Touch and Aberrant Mast Cell Accumulation in the Affected Tissues

Using standard conventions of species scaling practices, we used 10,000 ppm (1% *w*/*v* in saline; 100 times the safe human dose [22] of 100 ppm, i.e., ppm) as a sensitizing dose and a lower 0.5% dose for subsequent challenges of MI dissolved in saline in our experiments using 6–8-week-old outbred ND4 female mice. These doses are similar to those used for dermal sensitization and challenge using MI in CBA [25] mice, as well as dermal and airway sensitization and challenge with MI in C57BL/6 and BALB/C mouse strains [26]. To our knowledge, we are the first to use MI in ND4 mice and, therefore, we first confirmed that these sensitization and challenge doses caused detectable and significant ear-swelling responses in flank-sensitized ND4 mice after three topical challenges on the ear (Appendix A).

Next, we sensitized mice with 1% and 0.5% MI dissolved in saline on their shaved flanks before administering 10 daily challenges of 0.5% MI or saline in their vaginal canals (Figure 2A). We assessed changes in tissue mast cells after 10 intra-vaginal challenges with 0.5% MI or saline and found that one day after 10 MI challenges, there were 1.75 times as many mast cells in the vaginal canal tissue of sensitized female ND4 Swiss mice compared with vehicle-challenged controls, although this increase was no longer detectable by 21 days (Figure 3A–G). This was accompanied by a significant increase in serum IgE levels in MI-challenged mice one day after the cessation of challenges (Figure 3H); circulating IgE is important for mast cell expansion and survival [27,28]. Furthermore, we observed that sensitized female ND4 Swiss mice were more sensitive to touch, as measured with an electronic Von Frey meter, with a 60% decrease in withdrawal threshold one day after 10 exposures to MI in the vaginal canal (Figure 3I). Shaved and sensitized mice that were treated with 0.9% saline were significantly less sensitive than their MI-treated counterparts and did not display a similar decrease from baseline. MI-challenged mice remained significantly more sensitive than saline-treated controls for up to 14 days (Figure 3I). These observations of early mast cell accumulation, elevated serum IgE, and consequent painful sensitivity in response to MI exposures are congruent with similar outcomes we have previously described in mice exposed to commonly used laboratory haptens Ox and DNFB [7,8,9], and suggest that this ubiquitous household preservative can induce allergy-provoked pain.

### 2.3. Repeated Exposures to MI in the Vaginal Canal Induce Inflammatory Changes in the Vaginal Mucosa and in the Spinal Cord of Mice

In our previous experiments, we found that repeated intra-vaginal exposures to the hapten DNFB induced increased levels of *interleukin (IL)-6* and *interferon-γ (IFN-γ)* transcripts in the vaginal canal [9]. Here, we found that one day after the 10th MI challenge, average relative transcript abundance of mRNAs encoding *IFN-γ* and *IL-6* in the vaginal canal tissue of MI-treated ND4 Swiss mice increased twofold over vehicle-challenged mice (Figure 4A). However, at seven days following 10 MI challenges, the amount of *IFN-γ* and *IL-6* transcripts was comparable to vehicle-challenged controls (Figure 4A).

As painful responses to touch persisted for more than two weeks after the cessation of MI exposures in the vaginal canal, we next looked for changes in levels of immunomodulatory factors in the spinal cord of MI-treated versus control mice. IL-1 and IL-6 signaling in the spinal cord are associated with low grade chronic inflammation [29], and intra-thecal injection of anti-IL-6 antibodies has been shown to alleviate pathological pain [30]. We evaluated the relative transcript abundance of *IL-1β* and *IL-6* in the spinal cord of both MI-treated and control mice one and seven days after the 10th challenge. The levels of *IL-1β* were increased by 1.5-fold in the spinal cord one day after 10 challenges, but this increase was resolved by seven days after cessation of challenges (Figure 4B). *IL-6* transcripts were slightly elevated by 0.75-fold to 2-fold in the spinal cord of MI-treated mice at one day after 10 challenges, but the increase was also no longer detected at seven days.

Given the increase in eosinophils accompanying pain-inducing allergic exposures to hapten oxazolone we observed earlier [7], we measured the levels of eosinophil peroxidase (EPO; a tissue marker for activated eosinophils) in the vaginal canal after 10 exposures to MI. One day after both 3 and 10 MI challenges, the levels of EPO were significantly higher in MI-treated vaginal canals than in vaginal canals exposed to saline. One day after 10 challenges, mice had around three times higher levels of EPO in MI-treated vaginal canal tissue than controls (Figure 4C). Seven days after the 10th challenge, eosinophil activity in the tissue was no longer increased as compared with saline controls. Mice that were multiply exposed to oxazolone showed neutrophil influx into the painful, allergic tissue [7]. Here, we saw a slight, but not significant increase in myeloperoxidase activity, indicating neutrophil influx into MI-challenged tissue after 1–3 administrations of MI in the vaginal canals of sensitized mice; however, by one day after the 10th administration of MI, we could see no differences in myeloperoxidase activity between saline-challenged and MI-challenged mice (Appendix A).

### 2.4. Therapeutic Administration of THC in the Vaginal Canal after 10 MI Exposures Reduces Both Mast Cell Numbers as Well as Painful Sensitivity to Touch

Intra-vaginal therapeutic application of THC alleviated DNFB-provoked allergy-driven genital pain in our previous studies [9]. To assess any effects of therapeutic THC treatment following induction of MI-provoked mast cell increase and heightened tactile sensitivity, we administered 50 µg of THC intra-vaginally in 20 µl saline for six consecutive days in the vaginal canal beginning one day after the tenth MI challenge (Figure 2B). One day after six THC treatments, mast cell numbers decreased by 45% compared with mice that received no THC (Figure 5A–C). Seven days after THC treatment, mast cells were slightly decreased by 7% of those observed in untreated MI-challenged mice. By 21 days after the last THC treatment, mast cell numbers for both the no treatment (NT) and THC treatment groups were indistinguishable from one another (Figure 5B); this coincided with the resolution of mast cells to baseline levels after 10 MI challenges (Figure 3A–G). We evaluated tactile sensitivity in mice treated with therapeutic THC and found that, one day following cessation of THC treatment, tactile sensitivity was significantly reduced compared with untreated mice, with THC-treated mice displaying a ~20% decrease from baseline and mice not treated with THC displaying a 45% decrease (Figure 5D). By seven days after the last THC treatment, the sensitivity thresholds for mice treated with THC were at the same levels as mice that did not receive THC.

### 2.5. Preventive Administration of THC in the Vaginal Canal before and during 10 MI Exposures Reduces Both Mast Cell Numbers as Well as Painful Sensitivity to Touch

We next treated MI-challenged mice with THC before and during exposures to MI to evaluate the potential preventive effects of THC in MI-driven allergy-induced pain. We applied THC inside the vaginal canal 12 h before each MI challenge as well as once 12 h after the 10th challenge (Figure 2C). One day after the cessation of MI challenges, and 12 h after the last THC treatment, mast cell numbers in the THC-treated vaginal canal tissue showed a ~0.6-fold decrease in comparison with non-THC treated mice exposed to MI in the vaginal canal daily for 10 days (Figure 6A,C,D). This comparative decrease persisted at seven days after the last MI challenge; with avidin signal intensity of THC treated MI-challenged mice decreased by ~50% (Figure 6B,C,E). At 21 days after the 10th MI challenge, mast cell numbers between THC-treated and control MI-challenged mice were similar, reflecting the decrease in mast cell density seen at 21 days after 10 MI challenges. Importantly, preventive THC treatment resulted in a significant, persistent reduction in mean tactile sensitivity at one and seven days after cessation of treatment, compared with NT controls (Figure 6F). We observed decreases from the baseline of only 30% and 25% in mice treated preventively with THC at one and seven days after the cessation of MI challenges, respectively, compared with mice not treated with THC, who displayed a 55% decrease from the baseline at one day and a 50% decrease at seven days.

## 3. Discussion

Our collective understanding of risk factors, etiologies, and effective interventions for vulvodynia continues to evolve. Histories of seasonal and contact allergies [3], recurrent urinogenital infections [31], and chronic psychological stress [32,33] have all been positively associated with individual- and population-level vulvodynia risk. In our previously published work, we have demonstrated the biological plausibility of contact allergy reactions in labiar skin [7,8] and vaginal canal [9] tissues, driving subsequent persistent painful sensitivity to touch in the ano-genital ridge of ND4 female mice. We used hapten irritants, oxazolone and DNFB, commonly used in the laboratory, to study contact hypersensitivity in mice and found a common pattern in the pathogenesis of allergy-driven pain. Multiple exposures to haptens led to abnormal tissue mast cell accumulation and tactile pain. Mast cell depletion, either via intra-labiar secretagogue compound 48/80 [8] or intra-vaginal topical THC [9], led to a temporary alleviation of painful symptoms.

A recent study by Reed et al. [17] suggests that environmental exposure to household and work-related toxins may amplify vulvodynia risk. Accordingly, we demonstrate here that the biocide preservative MI triggers allergy-driven, mast cell-dependent pain responses in a mouse model of vulvodynia. MI is widely used in cosmetics and cleaners, and has been the target of recent scrutiny as a population-level sensitizing allergen [12] and a risk factor for skin and airway injury [12,14]. Our meta-analysis of patch-testing studies (Figure 1) demonstrates significant recent clinical and epidemiological attention paid to MI toxicity in Europe. In 2014, the European Commission’s Scientific Committee on Consumer Safety regulated MI out of leave-on cosmetic products and lowered the safe limit of MI in rinse-off cosmetics from 100 ppm to 15 ppm [34]. In contrast, MI is still widely used at 100 ppm in cosmetics in the United States [24].

A limited number of mouse studies of MI toxicity in the past have used sensitizing doses of 1–2% MI followed by 0.5% MI used for elicitation [25,26]. We found that 10 intra-vaginal exposures to 0.5% MI in previously sensitized ND4 female mice caused painful ano-genital sensitivity and vaginal mast cell expansion at one and seven days after cessation of MI challenges, accompanied by increased local transcription of *IFN-γ* and *IL-6* genes and higher levels of circulating IgE. As avidin binds to proteoglycans present in granules, we acknowledge that it is possible that some of the avidin staining we see in the vaginal canal after 10 consecutive MI challenges might be coming from eosinophils or possibly neutrophils present in the tissue. However, as we show in Appendix A and Figure 4, eosinophils and neutrophils (as measured by tissue peroxidase activity) are no longer increased in the vaginal canal tissue by the time 10 MI challenges have been administered. These outcomes of MI exposure are similar to what we have observed with labiar and vaginal exposures to common laboratory haptens [7,8,9]. Mast cells are present in murine [35] and human [36] vaginal canal and are known to participate in local inflammatory responses to parasites [37], bacteria [38], and yeast [39]. Mast cell increases have been reported in subsets of vulvodynia patients [5,6], but not in others [40]. Foster and colleagues have described increased levels of inflammatory cytokines in vulvar tissue [41,42], while Reed et al. have reported not finding such differences in the population they studied [43]. However, taken together, multiple lines of evidence suggest that dysregulated inflammation is a part of the pathobiology of vulvodynia.

Pathological inflammation in allergic diseases has been linked, in part, to the bacterial dysbiosis and disruption of the commensal microflora of the gut, lungs, and skin [44]. Altered gut microbiomes have been reported in fibromyalgia patients experiencing chronic widespread pain [45] and associated with visceral abdominal pain [46]. In a pilot study of the effects of MI on the mouse vaginal microbiota, we found that repeated exposures to MI as well as saline vehicle initially disrupt the vaginal microbiota in our mouse model (Appendix A). However, beta diversity analyses suggest that only MI challenges continue to disrupt the microbiota, while saline-treated control mice return toward baseline stability (Appendix A). This difference in compositional stability between MI and saline treatments is greatest at days 16–17 post-sensitization (Appendix A). We are currently conducting further experiments to monitor changes in both vaginal and gut microbiota after MI exposures and to investigate resulting dysregulated inflammation in the vaginal canal. Disruptions of this ecosystem have been linked to increased susceptibility to several diseases including bacterial vaginosis [47,48], chlamydia [49], and endometriosis [50], and thus are of interest to the larger pathological implications of this work. In our pilot studies, we see that after three exposures, MI-treated mice also had detectably higher levels of IL-1 and CXCL-2 in vaginal lysates and continued to have higher expression of the genes encoding these cytokines up to a week after MI challenges ended (Appendix A), suggesting continued heightened inflammation in the vaginal canal. The dynamics of the vaginal microbiome in health and disease [51] are relatively understudied and may be a fruitful direction to pursue in understanding the inflammatory correlates of vulvodynia. Given the role of spinal IL-6 in chronic inflammation and pathological pain [30], we tested inflammatory gene expression in the sacral spinal cord of MI-challenged mice and found slight increases in both *IL-6* and *IL-1β* transcripts one day after 10 challenges (Figure 4). IL-1β, released by spinal microglia, has been implicated in inflammatory pain that lasts beyond the presence of the noxious environmental stimulus that triggered it in the first place [52]; this is similar to hapten allergen-driven pain that persists well after antigen challenge in our past and current mouse studies. Systemic increases in nerve growth factor (NGF) have been reported in the serum of some vulvodynia patients [45], and central sensitization of the nervous system reported in others [53]. As such, cellular and molecular mechanisms underlying central sensitization and its contributions to vulvar pain need to be further elucidated to determine whether novel biomarkers and/or targeted therapies can be developed to help at least a subset of the population in need. Although a history of allergies is known to amplify vulvodynia risk, eosinophil levels and activity have not, to our knowledge, been studied in vulvodynia patients. Eosinophils are critical mediators of allergic responses as well as long-term esophageal and airway remodeling [54]. We see MI-driven early increases in eosinophil activity (Figure 3); while specific roles of eosinophils in pain pathologies remain to be characterized, eosinophilia accompanies many painful chronic conditions, including endometriosis [55]. Eosinophils may be another important cellular player to consider in the pathogenesis of vulvar pain.

Vulvodynia is a multifactorial condition and likely has different, potentially overlapping etiologies. Our pre-clinical findings and those from clinical studies affirm that identifying inflammatory biomarkers and cellular mechanisms of action for different etiologies is warranted and may lead to novel effective, targeted therapies. We have observed before [8,9], and see here again, that in tissue allergy-driven persistent pain, mast cell depletion can help relieve tactile sensitivity. Six daily doses of intra-vaginal administration of 50 µg THC after multiple MI exposures alleviates painful responses temporarily for up to seven days, along with a concomitant ~50% reduction in tissue mast cells (Figure 5). Additionally, if pre-sensitized mice are treated daily with THC beginning 12 h before the first MI challenge until 12 h after the last MI challenge, they do not develop painful ano-genital tactile sensitivity (Figure 6). Our results suggest that mast cells may play a role in the cannabinoid-mediated pain reduction of allergy-triggered genital pain in our model, although the effects of cannabinoid signaling on mast cell survival are not known. While topical mast cell granule stabilizer cromolyn has not been efficacious in relief of vulvodynia in the clinic [56], to our knowledge, mast cell depletion therapies have not been tested. Similarly, effects of cannabis-based medicines in vulvodynia pain relief have not been investigated and may well warrant a closer look.

The vaginal mucosa, while susceptible to noxious environmental stimuli, is also known to induce tolerance [57], and we have previously seen that 10 labiar exposures to the hapten DNFB produce longer lasting pain compared with 10 vaginal exposures [9]. While our findings here provide critical support for the idea that intra-vaginal MI exposures can provoke local inflammation and genital pain, human populations are exposed to MI through multiple routes, including skin and airways, and to cumulatively higher doses than the likely already unsafe limit of 100 ppm. The ubiquitous use of methylisothiazolinone in a wide range of consumer products raises the concern that repeated exposures to methylisothiazolinone from multiple sites could exceed the 100 ppm limit, and consequently increase the risk of sensitization in the population. To investigate this idea, we are currently conducting studies to determine whether cumulative exposure to methylisothiazolinone from multiple routes of exposure elicits sensitization as effectively as exposure through a single, localized route. From a regulatory perspective, it is also important to understand the biological threshold at which sensitization and pain occur. Accordingly, we are conducting dose-response studies to determine the doses required to elicit sensitization and pain. In the context of increasing scrutiny of MI toxicity in both the scientific [12] and popular [58] literature, our work here draws attention to a hitherto overlooked harmful outcome of MI exposure—its potential to contribute to allergy-driven persistent genital pain. While a potential link between methylisothiazolinone, mast cells, and vulvar pain may provide new tools for prevention and intervention for those living with vulvodynia, increased understanding of mechanistic connections between chemical exposures, allergies, and chronic pain may well be of even broader use and importance.

## 4. Materials and Methods

### 4.1. Meta-Analysis of MI Sensitization Studies

A total of 164 studies of MI-sensitization published from 1995 to 2019 were collected from PubMed.gov using “methylisothiazolinone AND (patch test)” as search terms. All articles were parsed with Python using xml.etree.ElementTree API to extract author names, title, publication year, and journal title. For each study in which patch testing was used to determine sensitization, testing locations, percent sensitivity to MI (ppm), and testing year(s) were extracted manually. Graphs were plotted using R. Scatter plots were generated using the gather() function from the tidyr package and ggplot() function from the ggplot2 package [59]; maps were generated with get_stamenmap() and ggmap() functions in the ggmap package [60].

### 4.2. Animal Usage

Six to twelve-week-old naive female ND4 Swiss mice (Harlan Laboratories, Indianapolis, IN, USA) were housed with a 12 h light/dark cycle and free access to food and water in Macalester College’s animal facilities. Mice were euthanized using 100% CO_2_ inhalation at predetermined timepoints. This study was performed in accordance with the Guide for the Care and Use of Laboratory Animals of the National Institutes of Health. All experimental protocols were approved by Macalester College’s Institutional Animal Care and Use Committee (IACUC B16Su2, approved on 1 July 2016).

### 4.3. MI Sensitization and Challenge

For MI challenges on ears, mice were anesthetized with isoflurane (Baxter Healthcare Corporation, Deerfield, IL, USA) and their backs were shaved on day 1. Sensitization was performed on the following day with 100 µL of 1% MI in a 4:1 mixture of acetone and olive oil (Sigma-Aldrich, St. Louis, MO, USA). On day 8, mice were sensitized a second time with 100 µL of 0.5% MI. Beginning on day 12, mice were challenged with 20 µL of 0.5% MI on each ear, 10 µL on each side, once every day at the same time for three days.

For vaginal canal challenges, mice were shaved on the flank on day 1 and sensitized on the flank with 100 µL of 1% methylisothiazolinone (MI) (Sigma-Aldrich, St. Louis, MO, USA) dissolved in 0.9% saline on the next day. Four days later, mice were sensitized again on the flank with 100 µL 0.5% MI in 0.9% saline (Figure 2). From day 12 onwards, mice were challenged with 20 µL of either 0.5% MI or vehicle (saline) in the vaginal canal. Mice were challenged daily at the same time for either three (Appendix A) or ten days (Figure 2).

### 4.4. Ear Edema Measurements

Ear thickness was measured using digital calipers (±0.1 mm; VWR, Radnor, PA, USA) one and two days prior to the start of the experiment to establish the baseline for each mouse. Measurements were repeated 6 and 24 h after the third challenge. The percent difference between the baseline and post-challenge values was calculated for each mouse and then for each experimental group.

### 4.5. THC Treatments

THC was precipitated from a methanol extract (Sigma-Aldrich, St. Louis, MO, USA) and resuspended in 0.9% saline with a few drops of olive oil (Sigma-Aldrich, St. Louis, MO, USA) to promote emulsification. Then, 50 μg of THC in 20 μL of saline solution was gently pipetted into the vaginal canals of mice. Therapeutic treatments were administered for six consecutive days after the 10th MI challenge (Figure 2B). Preventive treatments were administered daily starting 12 h prior to the first MI challenge and ending 12 h after the 10th MI challenge (Figure 2C).

### 4.6. Tissue Collection and Storage

Whole blood was collected immediately after euthanasia via heart puncture; cells were removed by centrifugation, serum aliquoted, and stored at –80 °C. Vaginal canal and spinal cord tissues were carefully extracted 1, 7, and 21 days after MI challenges or THC treatment, as indicated in the figures. Vaginal canals were carefully extracted, and the section between the cervix and introitus was used for further analyses. Sacral sections 1 and 2 of the spinal cord were identified as described [61], collected, and flushed with cold 1× phosphate buffered saline (PBS) (MidSci, St. Louis, MO, USA). All tissues extracted for semi-quantitative real-time reverse transcription PCR (sqRT-PCR) were weighed immediately after collection, flash-frozen, and stored at –80 °C. Tissues extracted for immunofluorescent staining were flash-frozen and stored at –80 °C. Tissues extracted for eosinophil peroxidase analyses were weighed after extraction, stored in 0.5% hexadecyl trimethylammonium bromide (HTAB) buffer at a volume 5.6 times the wet tissue mass, flash-frozen, and stored at –80 °C. Vaginal canal tissues used for protein quantification were flash-frozen and stored at –80 °C.

### 4.7. Tactile Sensitivity

Mechanical sensitivity was measured in a 2 × 2 mm area of the ano-genital ridge of mice with an electronic von Frey Anesthesiometer (IITC Corporation, Woodland Hills, CA, USA). Mice were allowed to acclimate in individual Plexiglass von Frey chambers over a wire mesh grating 15 min prior to any sensitivity measurements. Two baseline measurements were taken one and two days prior to MI sensitization; mice were screened for optimal responsiveness and sorted into experimental groups with comparable average baseline responses, as previously described [7,8,9]. Three to four experimental measurements were taken at described timepoints after 10 MI challenges and/or THC treatments by an experimenter blinded to the treatment groups. The averages of experimental measurements were calculated for each timepoint and reported as a percent decrease from the baseline. Percent decreases from the baseline of 33% or higher were considered hyperalgesic [7,8,62].

### 4.8. Immunofluorescent Staining and Microscopy

Flash-frozen vaginal canal samples collected from mice 1, 7, and 21 days after ten challenges were embedded in optimal cutting temperature compound (Sakara Finetek, Torrance, CA, USA) and 12 μm sections were cut using a Leica CM 1860 cryostat. Sections were fixed in 4% paraformaldehyde (Sigma-Alrich, St. Louis, MO, USA; pH 8.5), permeabilized for 30 min with 0.1% Triton (Sigma-Aldrich, St. Louis, MO, USA)/PBS, and blocked for one hour in 5% normal donkey serum/PBS. For mast cell staining, slides were incubated for one hour with Fluorescein Avidin D (Vector Laboratories; Burlingame, CA, USA) (1:1000) to stain polysaccharides in mast cell granules, as described by Kakurai et al. [63] and in our own previous studies [7,8,9]. For mast cell quantification after MI or saline challenges, all slides were coverslipped with Vectashield + DAPI (Vector Laboratories, Burlingame, CA). Sections were imaged using a Zeiss LSM 800 laser scanning confocal microscope. Composite images of ten optical 1 µm sections projected on the z-axis were analyzed using Zen2.1 software (Carl Zeiss AG, Oberkochen, Germany). Mast cell density was determined by fluorescent pixel intensity measurements, taken in four representative 5000 μm^2^ regions of interest from one section. Three sections per slide were quantified for three slides per mouse. For mast cell quantification after THC administration, four representative 5000 µm^2^ regions of interest, as well as one blank region of interest outside of the tissue, were measured for three sections per each slide. The average of the four representative sections was taken, subtracted by the blank region of interest, and then divided by 5000 µm^2^ to give a value of the average fluorescent intensity/µm^2^ for each section quantified. Three sections per slide were quantified per mouse.

### 4.9. RNA Isolation and Quantification of Gene Expression

Total RNA was extracted from flash-frozen vaginal canal or spinal cord tissues using the Total RNA Mini Kit (Midwest Scientific, St. Louis, MO, USA). RNA was eluted with RT-PCR grade water, and quantified using either a Nanodrop ND-1000 Spectrophotometer (Thermo Fisher Scientific, Wilmington, DE, USA) or NanoPhotometer NP80 (Implen, West Lake Village, CA, USA). mRNA was reverse-transcribed in a 2720 Thermal Cycler (Thermo Fisher Scientific) using Superscript III First-Strand Synthesis System (Thermo Fisher Scientific). Relative transcript abundance was determined by sqRT-PCR using TaqMan Gene Expression Assay Primer/Probe Sets: interleukin-6 (IL-6; Mm00446190_m1), interferon-γ (IFN-γ; Mm01168134_m1), chemokine C–X–C motif ligand (CXCL-2; Mm00436450_m1), interleukin-1 β (IL-1β; Mm00434228_m1), and MasterMix (Life Technologies) in a StepOnePlus Real-Time PCR System (Life Technologies). The results were normalized to the expression of housekeeping gene β-2-microglobulin (β2M; Mm00437764_m1) and then calculated as fold-expression over vehicle controls [64], following methods used in previous studies [7,8,9].

### 4.10. Protein Quantification

Protein concentrations in serum and in vaginal canal lysates were determined using enzyme-linked immunosorbent assay (ELISA). IgE concentration was measured using an IgE ELISA kit (Bethyl Laboratories, Montogomery, TX, USA). CXCL2 and IL-1β levels were quantified using cytokine specific ELISA kits (R&D Systems, Minneapolis, MN, USA) from whole cell lysates, as previously described [8,9]. Absorbances for all ELISAs were measured at 450 nm and 570 nm using a PowerWave XZ microplate spectrophotometer (Biotek Instruments, Winooski, VT, USA). The recorded optical density measurements (OD) were then used to determine the protein concentration for each sample from a standard curve. Total protein concentrations in all samples were determined using a Detergent Compatible Assay (Bio-Rad, Hercules, CA, USA) following the manufacturers’ directions. Total protein concentrations were used to normalize concentrations of target proteins derived from ELISA assays.

### 4.11. Quantification of Eosinophil Activity

Vaginal canals stored in hexadecyl trimethylammonium bromide (HTAB) buffer were homogenized into whole cell lysates after adding 0.5% HTAB at a volume of four times larger than the storage buffer. All samples were sonicated, freeze-thawed thrice, re-sonicated for optimal homogenization, and centrifuged to separate cellular debris, after which the resulting supernatant was incubated in a substrate solution containing 0.025% hydrogen peroxide, 16 mmol/L o-phenylenediamine, and PBS in a 96-well plate. After 30 min, absorbances were read at 490 nm. All measured optical densities (OD) were divided by the mass of wet tissue to obtain OD/g of wet tissue.

### 4.12. Vaginal 16S rRNA-Based Microbiome Profiling and Analysis

To investigate the effect of MI sensitization and challenge on the vaginal microbiota, we collected vaginal microbial biomass by lavage as previously described [8] using 100 µl of sterile PBS before sensitization (day 0), post-sensitization (day 6), and after 3 and 10 daily challenges of MI or saline (experiment days 7 and 14). Lavage samples were frozen immediately at –80 °C until processing. Genomic DNA was extracted by the University of Minnesota Genomics Center, followed by 16S rRNA gene amplification using Nextera library-compatible primers flanking the V3–V4 hypervariable regions in a dual-indexing protocol [65]. Libraries were verified with 16S qPCR and normalized based on molecular copy number, and then sequenced on an Illumina MiSeq with v3 reagents in 2 × 300 paired-end mode.

We used the quality control pipeline SHI7 [66] to stich paired reads, trim adaptors, and a quality filter with a trimming threshold of 32 and a mean quality score of 33. This yielded 1,671,150 reads that were then aligned to a custom database from the NCBI RefSeq 16S rRNA Targeted Loci Project (https://www.ncbi.nlm.nih.gov/refseq/targetedloci/) at 97% identity with the accelerated optimal gapped alignment engine BURST [66], run in CAPITALIST mode. OTUs present in less than 5% of samples, and samples with less than 200 counts, were dropped, leaving 33 samples for downstream analysis, with an average read count of 10,411 per sample. Initial diversity analyses were performed using QIIME2 v.2018.11 (https://qiime2.org/) [67], and further statistical tests and visualizations were performed in R v3.5.0 (R Foundation for Statistical Computing, Vienna, Austria; https://www.R-project.org/), using the packages ggplot2 and splinectomeR [68].

### 4.13. Statistical Analysis

Data were processed using Excel (Microsoft, Redmond, WA, USA) or FlowJo Software (FlowJo, Ashland, OR, USA) and graphed using PRISM 5.0 (GraphPad, San Diego, CA, USA). One-way analysis of variance (ANOVA), post hoc Tukey honest significant different (HSD) analyses, or unpaired Student’s *t*-test were run using JMP software (v. 10, SAS, Cary, NC, USA) to compare treatment groups at designated time points. Statistical significance will be defined as *p* < 0.05, *p* < 0.01, and *p* < 0.001 between the two treatment groups and indicated by *, **, and ***, respectively.

## Figures and Tables

**Figure 1 ijms-20-05361-f001:**
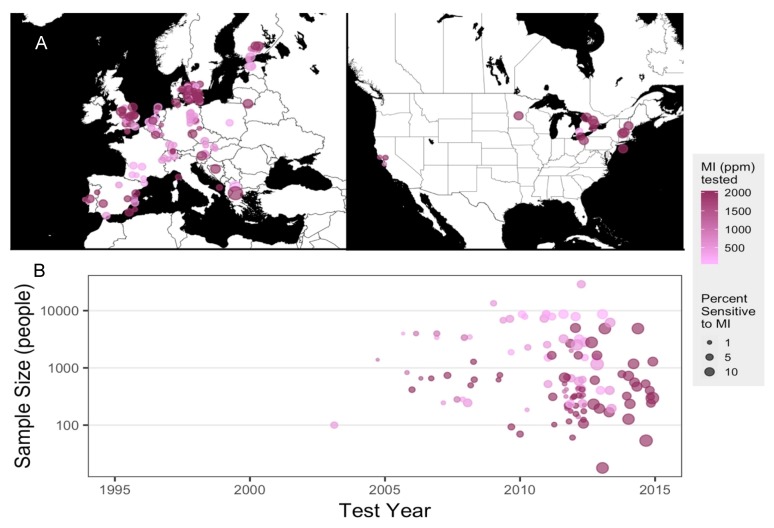
Population level sensitization to methylisothiazolinone (MI) in Europe and North America. (**A**) Location of epidemiological studies conducted in Europe (left) and North America (right); all points plotted at the approximate latitude and longitude of the original study with random noise added for easier visualization. (**B**) Sample size reported by epidemiological studies in Europe and North America using MI patch tests. Point placement corresponds with the year each study ended. Sample size may include multi-year studies if yearly data were not available. Color gradient indicates the highest concentration of MI in parts per million (ppm) tested in each study and the size of the point represents the proportion of the population that tested sensitive to MI. Studies in which MI ppm were not reported are excluded from the plot. The size of each point represents the percent of participants that were found to be sensitized to MI (*n* = 151 studies).

**Figure 2 ijms-20-05361-f002:**
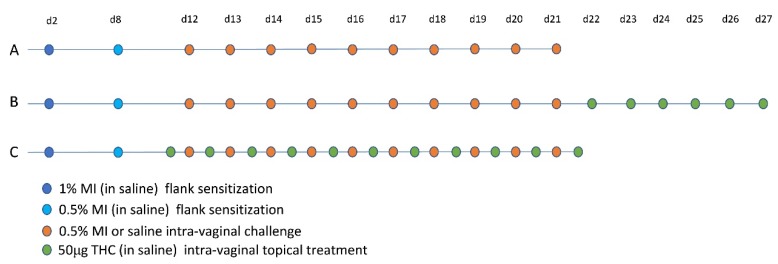
Sensitization, challenge, and treatment timelines. Schedule of MI in saline flank sensitizations and challenges (**A–C**). (**B**) Therapeutic intra-vaginal Δ-9-tetrahydrocannabinol (THC) treatment timeline. (**C**) Preventative intra-vaginal THC treatments.

**Figure 3 ijms-20-05361-f003:**
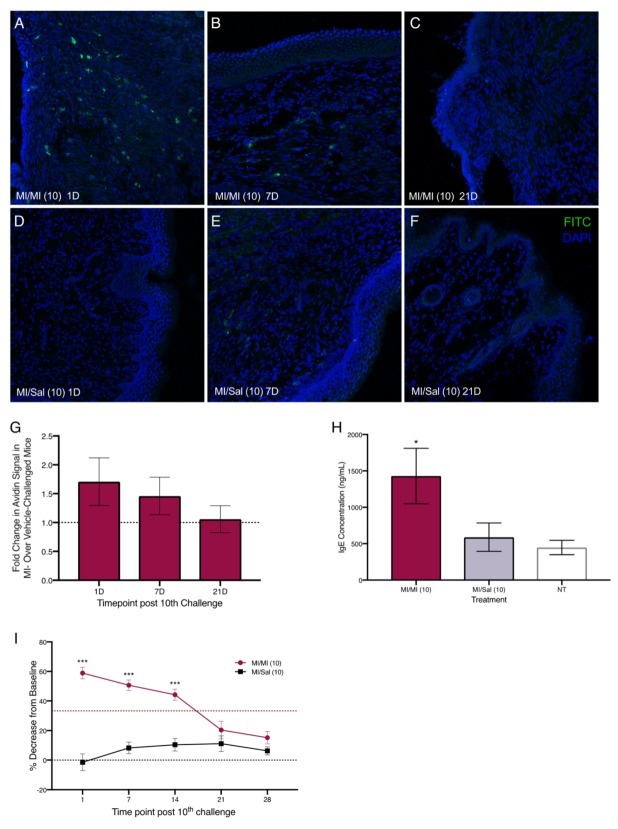
Increased mast cell density in the vaginal canal and elevated tactile ano-genital sensitivity after 10 intra-vaginal MI challenges in previously sensitized ND4 female mice. Representative confocal images of vaginal canal tissue from MI sensitized mice challenged with MI (**A**–**C**) or saline **D**–**F**) at 1, 7, and 21 days after the 10th MI challenge, respectively. Mast cells stained with FITC-conjugated avidin (green) and nuclei counterstained with DAPI (blue); 200× magnification. (**G**) Density of avidin^+^ mast cells in 12 μm vaginal canal cryo-sections from sensitized mice challenged with MI or saline. Results reported as fold change in avidin signal in MI- over saline-treated mice. Dotted line denotes no change. Data pooled from 5–6 mice. (**H**) Serum IgE content in mice treated with MI or saline in the vaginal canal 1 day after the last MI/saline challenge. NT bar denotes serum IgE levels in naïve age-matched, untreated mice. Significance with respect to vehicle control group * = *p* < 0.05; 4–6 mice/treatment group. (**I**) Tactile sensitivity in MI and saline challenged mice, reported as mean ± SEM of the percent decrease from baseline in the withdrawal threshold for each treatment group; *n* = 17–18 mice/treatment group. Red dotted line = 33% hyperalgesia threshold. Significance with respect to vehicle control group *** *p* < 0.001.

**Figure 4 ijms-20-05361-f004:**
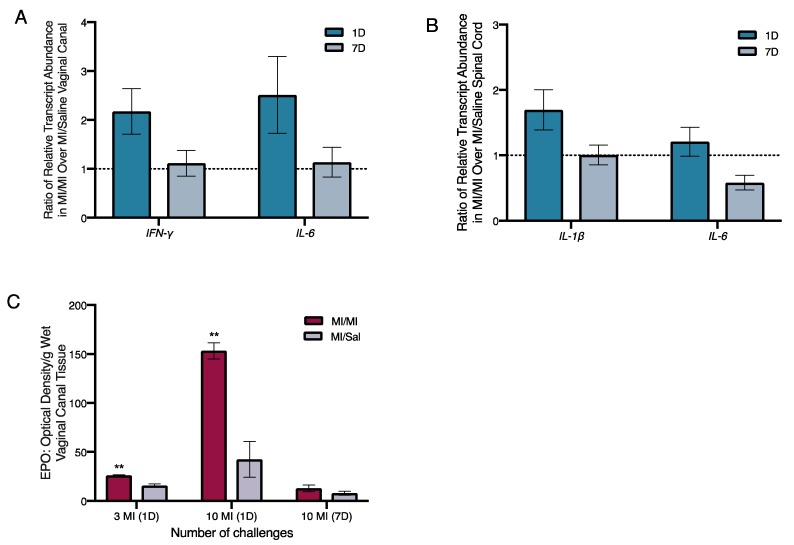
Inflammatory changes in the vaginal canal and spinal cord, and increased eosinophil activity after 10 intra-vaginal MI challenges in previously sensitized ND4 female mice. Relative transcript abundance of *interferon-γ (IFN-γ)* and *interleukin (IL)-6* transcripts in the vaginal canal tissue (**A**) and *IL-1β* and *IL-6* in spinal cord tissue (**B**) of MI challenged mice one and seven days after 10 challenges, normalized to *β2-microglobulin* mRNA levels; 5–6 mice/treatment group. Black dotted line denotes no change in the relative abundance of transcripts. (**C**) Tissue eosinophil peroxidase levels measured by optical density (OD)/g of wet tissue, in the vaginal canal of mice one day after the 3rd and 10th MI challenge; 3–5 mice/treatment group. Significance with respect to vehicle control group ** *p* < 0.01. EPO, eosinophil peroxidase.

**Figure 5 ijms-20-05361-f005:**
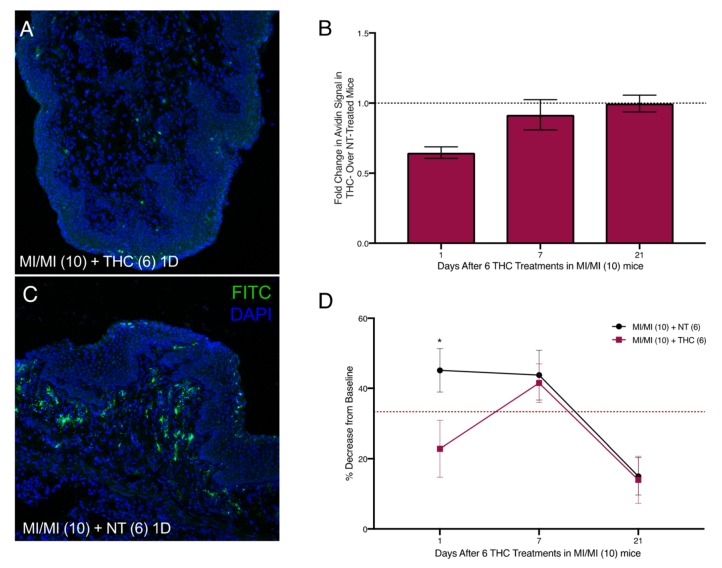
Reduced vaginal mast cell density and ano-genital sensitivity following six intra-vaginal therapeutic THC treatments after 10 vaginal MI challenges in previously sensitized ND4 female mice. (**A**,**C**) Representative confocal images of vaginal canal tissue from mice that were sensitized and challenged with MI and subsequently treated with THC (**A**) or untreated, that is, NT (**C**), at one day after the last treatment, respectively. Mast cells stained with FITC-conjugated avidin (green) and nuclei counterstained with DAPI (blue); 200× magnification. (**B**) Mast cell density displayed as fold change in avidin signal in THC-treated over NT mice at 1, 7, and 10 days after the 6th THC treatment; 6–7 mice/treatment group. Black dotted line denotes no change in MC abundance. (**D**) Anogenital tactile sensitivity of MI challenged mice treated with NT (black) or therapeutic THC (red) 1, 7, and 10 days after the 6th THC treatment; *n* = 8–9 mice/treatment group. Results displayed as mean ± SEM. Significance with respect to control group * *p* < 0.05.

**Figure 6 ijms-20-05361-f006:**
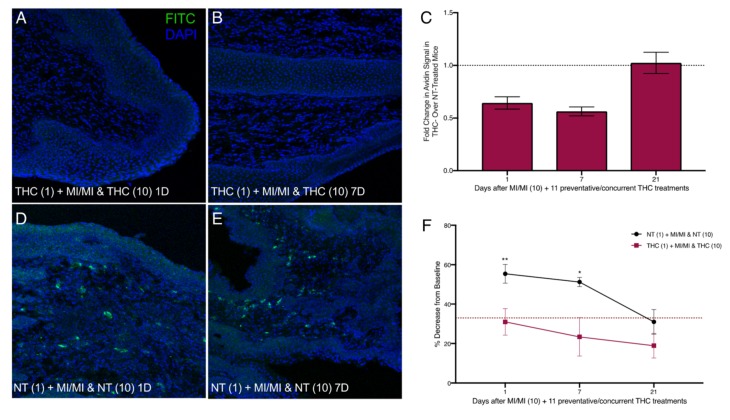
Reduced vaginal mast cell density and ano-genital sensitivity following intra-vaginal preventive THC treatments before and during 10 vaginal MI challenges in previously sensitized ND4 female mice. (**A**,**B**,**D**,**E**) Representative images of vaginal canal tissue from THC-treated (**A**,**B**) and NT (**D**,**E**) MI-challenged mice one and seven days after the 10th MI challenge. Mast cells stained with FITC-conjugated avidin (green) and nuclei counterstained with DAPI (blue); 200× magnification. (**C**) Mast cell density displayed as fold change in avidin^+^ signal intensity of the THC-treated over NT mice; *n* = 3–7 mice/treatment group. (**F**) Ano-genital tactile sensitivity of preventive THC-treated and NT mice at 1, 7, and 21 days after the 10th MI challenge; 8–9 mice/treatment group. Red dotted line denotes 33% hyperalgesia threshold. Results displayed as mean ± SEM. Significance with respect to control group * *p* < 0.05 and ** *p* < 0.01.

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
