# Peer review of "Repeated Vaginal Exposures to the Common Cosmetic and Household Preservative Methylisothiazolinone Induce Persistent, Mast Cell-Dependent Genital Pain in ND4 Mice"

_ijms, 2019, doi:10.3390/ijms20215361_

Round 1

Reviewer 1 Report

The  premise of these studies is sound and very interesting and would add to the current literature. The effects of pesticides on reproduction and in this case feminine intimacy is of great interest.  The authors have done an excellent review of the recent toxicology literature.  This paper is original, the content’s significance is high and the interest to readers would be high also.  The use of THC to mediate pain and  proposed effects on mast cell is novel.   There are flaws in the quality of the presentation of some of the major data generated which are the basis of this paper.  The method the authors used for  visualizing mast cells is not acceptable.  The authors use avidin  which in the 1980’s was acceptable.  The scientific community has learned that avidin will non-specifically bind polysaccharides on the surface of numerous cells types.  The authors have only shown low magnification photomicrographs of the vaginal tissue with no high power magnifications insets of the cells of interest; thereby to  not proving what the cells are.  Toluidine blue is presently the generally accepted method for visualizing mast cells. Though laborious toluidine blue is used by many laboratories. Through the use of computer programs like Image J granulated and degranulated masts can be enumerated.  An exciting addition to the paper would be the determination of the effects of THC on mast cell degranulation (histamine release).  The authors need to add hematoxylin and eosin stained tissue sections to show the histology/pathology of the vaginal tissue.  This will help the readers to orient themselves and show the inflammatory cell infiltrate. Major revisions are necessary and must include the addition of light microscopy of the tissue sections using H&E and toluidine blue staining.  There must be both low magnification photomicrographs  of tissue sections with insets of the cells of interest.  Therefore, though this paper is novel in its present state it must be rejected. 

Reviewer 2 Report

This is a very carefully performed research that includes both a meta-analysis of chemical (MI) sensitization reports across Europe and mouse studies to investigate the cellular basis of the sensitization. The authors demonstrate using mouse models that mast cells are early responders to treatment of mice with MI and that a neuroprotective agent (HT) reduces both mast cell number and pain in the mice. These observation link chemical sensitization to neuronal activation of mast cells , inflammatino, and sensation of pain. It is not clear from the study which comes first, mast cells activating the nerve cells or vise versa. The report raises questions that need to be studies in the future.

A minor point is that the authors  mention bacteria in their Discussion and possible causes, but this may have nothing to do with chemical sensitization, and they do not present any data on bacteria either.
